# Mapping Carbon Monoxide Pollution of Residential Areas in a Polish City

**Janusz Kwiecień * and Kinga Szopińska** 

Faculty of Civil and Environmental Engineering and Architecture, UTP University of Science and Technology, 85-789 Bydgoszcz, Poland; Kinga.Szopinska@utp.edu.pl

*   Correspondence: jkw@utp.edu.pl

**Abstract:** Road traffic is among the main sources of atmospheric pollution in cities. Maps of pollutants are based on geostatistical models using a digital model of the city along with traffic parameters allowing for ongoing analyses and prediction of the condition of the environment. The aim of the work was to determine the size of areas at risk of carbon monoxide pollution derived from road traffic along with determining the number of inhabitants exposed to excessive CO levels using geostatistical modeling on the example of the city of Bydgoszcz, a city in the northern part of Poland. The COPERT STREET LEVEL program was used to calculate CO emissions. Next, based on geostatistical modelling, a prediction map of CO pollution (kg/year) was generated, along with determining the level of CO concentration ($mg/m^3$/year). The studies accounted for the variability of road sources as well as the spatial structure of the terrain. The results are presented for the city as well as divided into individual housing estates. The level of total carbon monoxide concentration for the city was 5.18 $mg/m^3$/year, indicating good air quality. Detailed calculation analyses showed that the level of air pollution with CO varies in the individual housing estates, ranging from 0.08 to 35.70 $mg/m^3$/year. Out of the 51 studied residential estates, the limit value was exceeded in 10, with 45% of the population at risk of poor air quality. The obtained results indicate that only detailed monitoring of the level of pollution can provide us with reliable information on air quality. The results also show in what way geostatistical tools can be used to map the spatial variability of air pollution in a city. The obtained spatial details can be used to improve estimated concentration based on interpolation between direct observation and prediction models.

**Keywords:** air quality; carbon monoxide; COPERT STREET LEVEL; geostatistical modeling; residential area

## 1. Introduction

One of the main problems influencing the environment and health of people is the quality of air [1] (pp. 103–105) and [2–9]. The dynamic increase in the number of vehicles travelling on European roads, which is accompanied by a significantly slower development of the transportation network [10], has led to a noticeable drop in the average speed at which vehicles travel. This leads to several problems where the urban environment is concerned, including increased air pollution [11], which may result in more frequent occurrence of respiratory problems in inhabitants [12]. Road vehicles emit various types of greenhouse gases and air pollutants [13] (pp. 17–19). These pollutants can be divided into two groups: those which are regulated by the European Union road transport law and those which are not currently regulated [14]. The group regulated by provisions of European directives includes carbon monoxide (CO) alongside such pollutants as: carbon dioxide ($CO_2$), hydrocarbons (HCs), particulate matter (PM), and nitrogen oxides ($NO_X$). CO is a product of incomplete combustion, which occurs when carbon in fuel is only partially oxidized. Its increase in cities is observed especially

during rush hour traffic [8]. The compound is clear, odorless, and extremely toxic, which is why it is referred to as the "silent killer" [15,16]. Direct exposure to CO decreases oxygen flow in blood and is particularly a threat to people suffering from heart problems, as well as contributing to the formation of ozone and smog in lay directly above the ground surface [17]. In the recent years, the amount of carbon monoxide emitted from engines has decreased [8], among others by applying three-way catalytic converters [18], though the problem of CO emission from motor vehicles continues to exist. Despite the growing problem of smog, there is a lack of detailed studies on the emission of pollutants generated by motor vehicles on air quality, which attention had been drawn to in the works [10,11].

The Directive 2008/50/EC of the European Parliament and of the Council of 21 May 2008 on ambient air quality and cleaner air for Europe binds every member state to take measures connected with environmental protection, as well as emphasizing the need to limit pollution to levels minimizing its harmful effects on human health [14]. In accordance with the provisions of the directive, the limit level of carbon monoxide cannot exceed 10 mg/m$^3$/year, at a 60% range of tolerance. The margin of tolerance is understood as the value by which exceeding the limit levels of a substance in the air does not lead to the obligation of preparing an air protection program. At the same time, the limit value of CO increased by the range of tolerance is considered to be 16 mg/m$^3$/year. In all EU countries, the assessment of air quality is carried out in accordance with the procedure provided in the directive [14], and referred to a given zone. A zone is a city with over 100,000 inhabitants. For these zones, the level of harmful substances is determined, including the exceedance of limit levels, target levels, long-term target levels as well as the ranges of tolerance. Moreover, the assessment of the quality of air in zones is carried out with division into three classes: Class A in which the level of substances does not exceed the limit level or target level; Class B in which the level of at least one substance falls between the limit level and level increased by the margin of tolerance; Class C in which the level of at least one substance exceeds the limit level increased by the margin of tolerance or target level. Zones lying in Class C are regarded as areas of bad air quality (CO > 16 mg/m$^3$/year) for which repair actions ought to be assumed, aiming to improve the quality of air by preparing an air protection program of the zone. Maps of pollutants are the main diagnostic tools applied by authorities in an effort to solve the growing problem of bad air quality in cities (zones), as well as plan preventive and mitigation and remediation measures to be taken [19]. They present the average level of pollution in a given time interval. Currently, these maps are created based on measurement data taken from sensors, the amount of which is insufficient to fully recognize the level of pollution in cities [8]. Moreover, the use of data registered by a network of monitoring stations leads to modelling uncertainty [20] and requires integrated information provided by deterministic models based on the meteorology of emissions as well as the chemical and physical characteristics of the atmospheric air [21–23]. Accounting for only data from measurement stations makes the detailed assessment of the level of pollution of air in the city (zone) difficult.

Prediction of carbon monoxide air pollution and mapping its level ought to be carried out using geostatistical methods in space and time [10]. A digital model of the city along with traffic parameters [24] (pp. 15–20), atmospheric models, field measurements, and remote sensing data [25] (pp. 15–17) ought to be used for prediction. The use of satellite remote sensing (e.g., the Copernicus Sentinel-5P satellite placed in Earth orbit on 13 October 2017 [26]) allows us to obtain rough data on air pollution in the form of trace gases, such as carbon monoxide, nitrogen dioxide, and ozone with a resolution of 7 × 3.5 km. This enables a preliminary assessment of air pollution in individual cities. Understanding the phenomenon is a topical challenge in the field of data quality assessment. Geostatistical tools solve various spatial problems [27–29] and allow for estimating the prediction error on the basis of the propagation of uncertainty. The potential for using 3D data to assess the quality of air was noticed by a research team from Krakow [30], who, based on data from the Integrated Spatial Data Monitoring System for the Improvement of Air Quality in Krakow, confirmed that geostatistical data can be successfully used for the assessment of air quality in a highly urbanized city. A modeling system comprising a road traffic model, a road transport emission model, and a computational fluid dynamics air quality model was used to assess air quality in a Portuguese City [11]. The level of air pollution

with nitrogen oxides (NOx) and carbon dioxide ($CO_2$) was determined in the work. The influence of applying various road traffic emission models on the precision of air quality modeling (for NOx as well as PM10 pollutants) at street level resolution was studied by Vicente and his research team [10]. The authors analyzed two emission models with different complexity levels regarding the ability to characterize traffic dynamics (transport emission model for line sources and vehicle-specific power model). Moreover, the model of pollutant dispersion in the atmosphere under diverse wind conditions was applied to carry out the simulation of air quality [10]. When modelling air pollution, in addition to geostatistical models, it is necessary to make use of special software using mathematical formulas to calculate emission levels of pollutants. Currently there are many such programs available, one of them being the COPERT program (computer program for calculating emissions derived from road transport) [31], which is the standard emission calculator in the EU. Implementation of the COPERT program is coordinated by the European Environmental Agency within the framework of activities of the European Topic Centre on Air pollution. The European Commission's Science and Knowledge Service manages the scientific development of the model. The COPERT package was developed with the aim of officially preparing a list of road transport emission in the European Economic Area member states. It can also be applied in experimental and implementation research. One of the modules of the package is COPERT STREET LEVEL [31], which is a new approach to calculating emissions from road transport.

The aim of the work was to determine the size of areas at risk of carbon monoxide (CO) pollution along with determining the number of people exposed to excessive levels of CO using a geostatistical model. The studies were carried out within the city of Bydgoszcz, located in the northern part of Poland. In the work, an analysis of the amount of carbon monoxide emitted by road traffic in 2018 on individual road segments was carried out with the use of the COPERT STREET LEVEL program. Next, based on the Kriging Ordinary Model, prediction maps of CO levels (kg/year) were generated for the entire city, as well as the levels of CO concentration [$mg/m^3$] for all the housing estates. For estates where limit levels of CO concentrations were found to have been exceeded, detailed analyses were carried out. Their aim was to assess the number of inhabitants exposed to excessive levels of carbon monoxide. Data derived from Bydgoszcz City Council as of the IV Quarter of 2018 were used in the study. The results of the studies are discussed in the last chapter. Despite the fact that the level of carbon monoxide concertation for the entire city was merely 5.18 $mg/m^3$/year, detailed studies revealed that, out of the 51 analyzed residential housing estates, ten are at risk of excessive levels of carbon monoxide. Moreover, the detailed studies showed that, in the area of the selected estates, nearly 45% of the population resides in areas with poor air quality. The above suggests that only detailed monitoring of air pollution, accounting for the variability of sources and spatial structure of terrain, can provide a reliable result of the assessment of the air quality in a city.

## 2. Materials and Methods

### 2.1. Research Area

The studies were performed in the city of Bydgoszcz, which is located in the north-central part of Poland. The city is the capital of Kujawsko-Pomorskie Province (Figure 1a). The number of inhabitants is 350,200, and when including the entire urban agglomeration, the number is over 470,000 inhabitants (as of 31 December 2018) [32], making it the eighth biggest city in Poland in terms of population. Bydgoszcz is an urban settlement with national significance, measuring 173.25 $km^2$ in surface area. Over 13% of the surface area of the city are residential areas, comprising 51 housing estates (which includes single family housing (7.28%), multi-family housing (5.09%), and residential and service buildings in the central zone of the city (0.37%)). As much as 47% of Bydgoszcz are green areas as well as surface waters, making the city very attractive to inhabitants [33]. In the city, the main types of economic activity include energy, electromechanical, chemical, and food processing plants, which are a potential source of carbon monoxide emissions [33]. Due to the terrain conditions and

transport services, the concentration of large business entities occurs in the peripheral areas of the city and reduces their emission nuisance (Figure 1b). For this reason, the main source of carbon monoxide emissions in the city is the use of coal or wood-fired stoves in single-family buildings and transport. Bydgoszcz is a specific city in terms the road layout. The development of urban space and the road network layout are characterized by a concentric nature. They are characterized by significant expansion in the east–west direction and a relatively small distance between the northern and southern border of the city. Bydgoszcz is an important road node in the northern part of Poland, being situated on the intersection of express roads No. S5 and S10 (Figure 1a). The high economic importance of Bydgoszcz, as well as its location at the intersection of main transportation routes, leads to a high share of heavy vehicles in the traffic structure. Due to the fragmentary ring road, transit transport also makes use of national roads lying in the center of the city (transit traffic moving through the city makes for over 13%). This problem pertains mainly to heavy goods vehicles moving in the north–south direction (Gdańsk–Poznań direction, national road No. 5) as well as transit in the east–west direction, which was partially assumed by the south-west ring road of the city (national road No. 10 as well as a fragment of the express road S10 (Figure 1b). Increased transit traffic in the city, caused by the lack of a complete ring road, significantly influences traffic difficulties as well as the formation of traffic jams caused by insufficient road capacity. There are two CO air quality monitoring stations in Bydgoszcz. They are located in the central part of the city, on the Śródmieście estate (Figure 1c). The first MS1 station was opened in 2010 and is located in the northern part of the estate, at Pl. Poznański Street. The second one was opened in 2013 and is located in the southern part of the settlement, at Warszawska Street. Measurements made at the stations are based on the annual air quality assessment in the city [33].

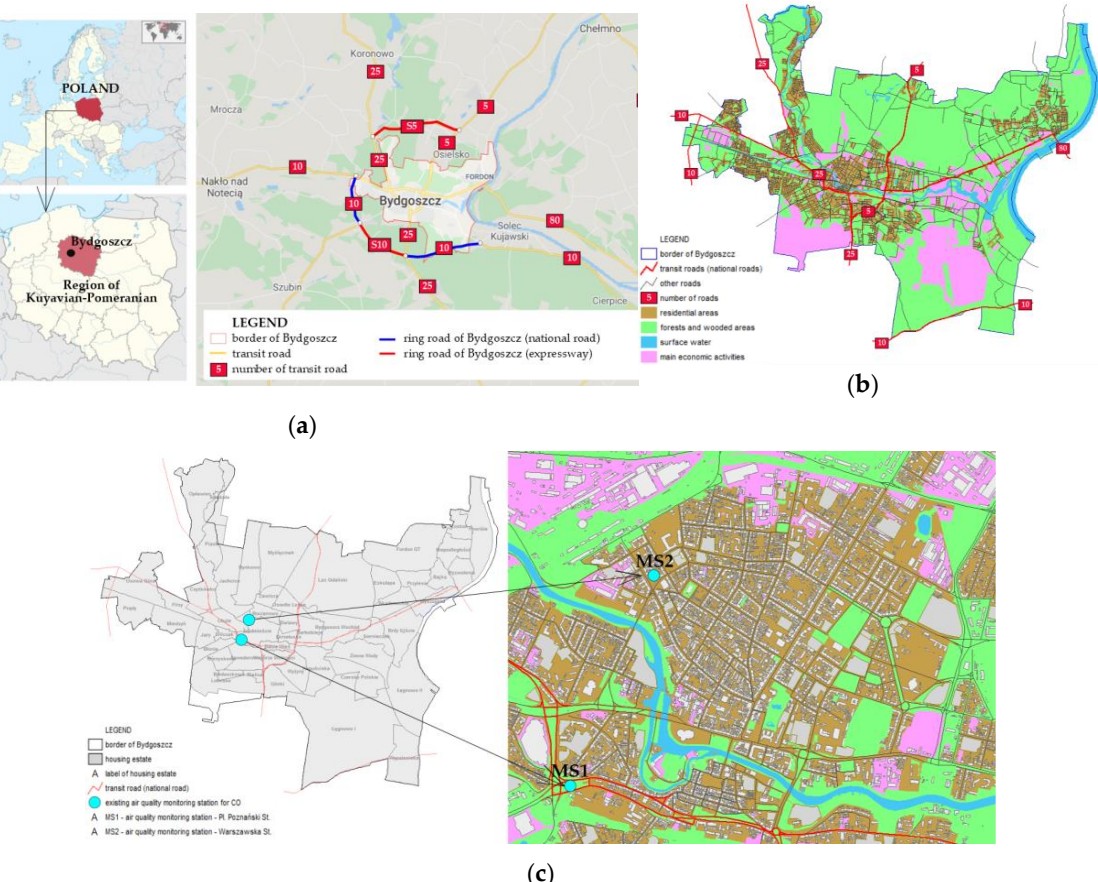

**Figure 1.** (**a**) Location of Bydgoszcz against the background of maps of Europe and Poland, along with the existing transit roads; (**b**) spatial structure of Bydgoszcz with marked roads; (**c**) location of the current air quality monitoring stations for CO in the Śródmieście housing estate in Bydgoszcz.

## 2.2. Research Methodology and Data

The methodology involved the following steps (Figure 2):

- determining the average daily traffic in the annual average daily traffic (AADT) on each section of road (link);
- calculating carbon monoxide emission with the use of COPERT STREET LEVEL (CSL) for every road section;
- determining the spatial relationship in a non-sample location in the form of a so-called variogram and adjusting it to the theoretical model;
- generating a prediction map of CO (kg/year) pollution with the geostatistical method for the entire city (zone);
- calculating the total CO concentration (mg/m$^3$/year) for the city (zone);
- calculating the average CO concentration (mg/m$^3$/year) for housing estates;
- selecting the estates with an exceeded concentration of CO (mg/m$^3$/year);
- for the selected estates, determining the areas found in the two zones of carbon monoxide;
- for the selected estates, the selection of areas lying in the two zones of carbon monoxide pollution, including the areas outside the danger zone of carbon monoxide pollution—below 16 mg/m$^3$/year—as well as areas within the zone of carbon monoxide pollution—above 16 mg/m$^3$/year;
- for the selected estates, the determination of the number of people subjected to excessive levels of carbon monoxide.

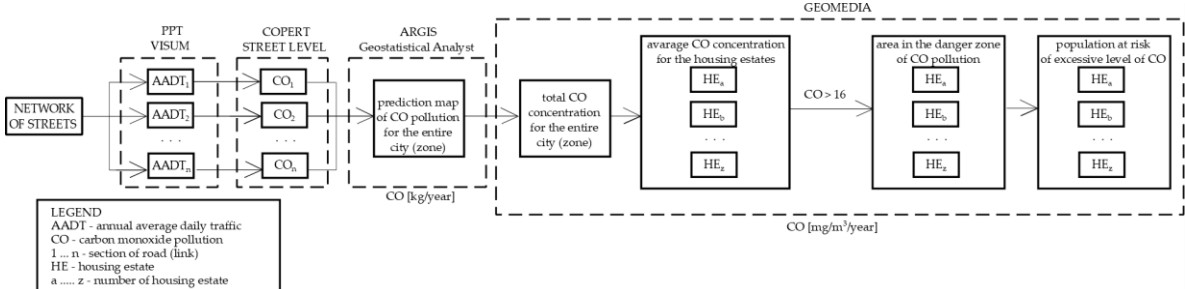

**Figure 2.** Stages of carbon monoxide pollution mapping using geostatistical modeling.

The first stage of the studies was determining the average annual daily traffic volumes (AADT) in the road transport (n/h) in the given road segment of the analyzing road [34]. AADT was the basic parameter calculated for all segments of the road network, while the method of its calculation depends on the type of measurement segment [35] and [36] (pp. 80–85). AADT is defined as the number of motor vehicles moving along the give road segment in a 24-hour period, on average over the period of one year. Two types of measurement segments were considered: type P—basic segments on which direct measurements of traffic are taken during all hours—as well as type T—roads segments on which direct measurement is not taken (data can be obtained indirectly from the PPT Visum software [34]). In the case of segments P and T, the value of AADT ought to be calculated in accordance with the formula [34]:

$$AADT = \frac{M_R{\cdot}N_1 + 0.85 M_R{\cdot}N_2 + M_N{\cdot}N_3}{N} + R_N, \tag{1}$$

where: $M_R$ is the average daily traffic on weekdays (from Monday to Friday, between 6:00 and 22:00), 0.85 $M_R$ is the average daily traffic on Saturdays and pre-holiday days (between the hours 6:00 and 22:00), $M_N$ is the average daily traffic on Sundays and holidays, $R_N$ is the average night traffic (between the hours of 22:00 and 6:00), $N_1$ is the number of workdays in a year, $N_2$ is the number of Saturdays and pre-holiday days in a year, $N_3$ is the number of Sundays and holidays in a year, and $N$ is the number of all days in a year.

The analyses made use of data derived from the PPT Visum program. The results of measurements of traffic were contained in the GIS (Geographic Information System) database pertaining to road traffic in Bydgoszcz, containing 2745 segments (links). The geometry of the road network was connected with descriptive data, including information on the names of streets, types of motor vehicles, limit speed, and annual average daily traffic.

The COPERT STREET LEVEL (CSL) software model was used to calculate the level of pollution with carbon monoxide. The model calculates emissions from traffic at street level and can be applied to approximately calculate various air pollutants, including CO. The program is fully optimized, making it possible to visualize data on a GIS map and makes use of road networks connected with the road traffic database, with AADT being an essential parameter for calculating the emission of air pollutants. In CSL, the smallest road segment is called a link. The number of links has to be at least equal to the number of intersections, and all changes in the road network, such as curves, ought to be represented by separate links. The length of a link has to be sufficient to enable all vehicles to be accounted for during the construction of the model [37]. At the same time, each link contains descriptive attributes regarding the length of the link, the number of vehicles per hour, the speed of a vehicle, and the characteristics of a link (e.g., one-way or two-way street, number of lanes, etc.). Examples of carbon monoxide calculation have been presented in Figure 3 as well as a view of data imported from the Excel file.

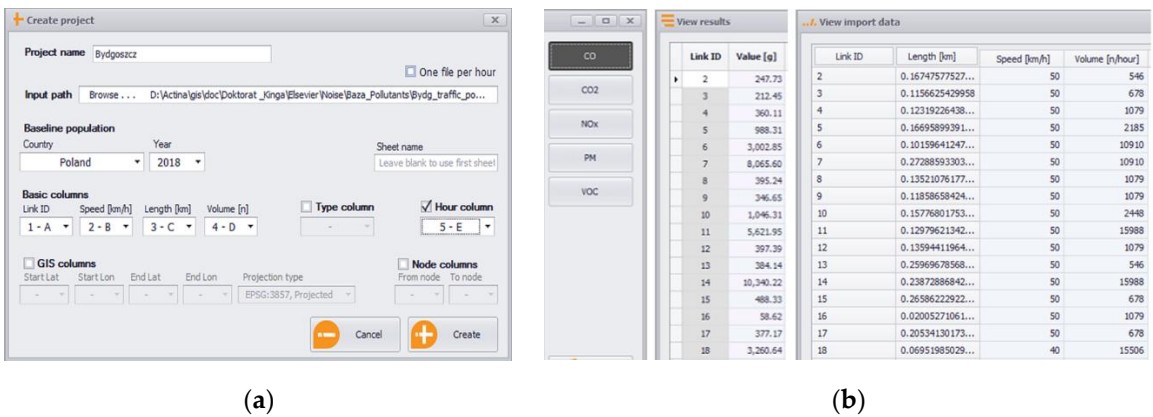

|    (a)    |    (b)    |

**Figure 3.** The COPERT STREET LEVEL (CSL) program: (**a**) main dialogue window; (**b**) fragment of results of calculations for carbon monoxide.

Environmental features, such as air pollution, are the result of many interacting physical, chemical, and biological processes. These processes are physically defined, although their interactions are complicated and subjected to random processes. Their complexity makes it impossible to make use of available deterministic or mathematical solutions. Currently, the best-known geostatistical method for modeling of this type of physical phenomena is kriging. There are many types of kriging [38,39] (pp. 37–47), or [40]. Each type determines the linear limitations of weights resulting from the condition of impartiality. In order to predict the phenomena in non-sampled locations, we must determine the spatial relationship. The geostatistical tool describing this relationship is referred to as a variogram on which we compare the sample values from a pair of points on the basis of the following equation [41]:

$$\frac{(z(x+h)-z(x))^2}{2}, \tag{2}$$

where: $x = \begin{pmatrix} x_1 \\ x_2 \end{pmatrix}$ is the vector of coordinate points in 2D, $h$ is the vector separating two points. A theoretical variogram (alternatively referred to as a semi-variogram) representing the average of square increases for interval h and is determined using the following equation [41]:

$$\gamma(h) = \frac{1}{2}E\Big[Z((x+h) - Z(x))^2\Big] \tag{3}$$

The variogram contains the following properties: zero at the origin $\gamma(0) = 0$, positive values $\gamma(h) \geq 0$, as well as the even function $\gamma(h) = \gamma(-h)$. Matching the values of the theoretical semi-variogram in the function of distance $h$ and experimental semi-variogram on the basis of data pertaining to CO values was carried out with the use of the ArcGIS Geostatistical Analyst program (Figure 4).

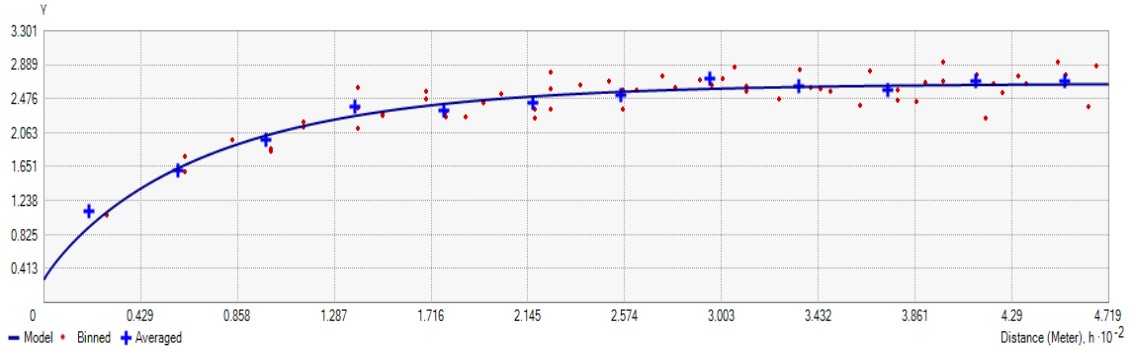

**Figure 4.** Semi-variogram of CO pollution.

After matching the model to the data, the modeling of CO concentrations for the entire city ought to be assumed. ArcGIS Geostatistical Analyst tools were used in the study. A point layer was generated every 10 m for each segment of the road network in which one of the descriptive attributes was the CO value determined in the CSL program. Prediction was carried out according for the following criteria [42]:

- prediction is based on the normalized root-mean-square error, which ought to be close to 1;
- prediction should not diverge significantly from the average value of the normalized standard error, the value of which ought to be as low as possible.

The exposure of people to car exhaust can be analyzed on three distance scales: near the source of emission (0.0–0.2 km), for an urban agglomeration (0.2–20 km), and for large regions (20–200 km) [43]. Modelling the scattered pollutants induced by motor vehicle traffic in an urban agglomeration is limited by the lack of observations of wind, contradictory to the hourly measurements low above the ground surface and measurements taken twice during the day in the upper atmosphere at the meteorological stations sometimes lying as far as hundreds of km from each other. Many models of vertical dispersion in urban areas are based on the estimation of the height (height at which the even mixing of atmospheric air occurs). For Bydgoszcz, the computational height assumed for calculating CO concentration levels in mg/m$^3$/year was 4 m, which results from information contained in the literature [43]. In order to calculate CO in the air, the volume of the urban space was calculated, assuming the value of the surface area in km$^2$ and height of dispersion in the vertical direction as 0.004 km. In connection with this, the formula for calculating the average annual CO production rate in mg/m$^3$/year was [44]:

$$average\ annual\ CO\ production\ rate = \frac{account\ of\ CO\left(\frac{kg}{year}\right)}{area\ value\ (m^3)}10^6 \tag{4}$$

The average level of CO concentration for individual estates was calculated using the Intergraph GeoMedia program. First, geometric merging of the point layer was carried out for each road segment in a given housing estate, and next, based on the functional attributes, the amount of CO concentration was calculated for the merged segments.

## 3. Results

First, based on data from the CSL program, a prediction map of carbon monoxide pollution in the city of Bydgoszcz was generated. As can be seen in Figure 5a, the amount of CO (kg/year) generated by individual road segments varies and depends on the traffic intensity. Based on the results presented in the prediction map (Figure 5b), i.e., the average square prediction error of 0.7357157 as well as average value of normalized standard error amounting to 0.07962313, it can be stated that the applied model fulfills the criteria given in Section 2.2. Analyzing the spread of carbon monoxide in the area of the city, it can be stated that the greatest amount concentrates around roads (ranging from 0.29–0.52 kg/year), with the level falling along with increasing distance to reach values in the range of 0.0001–0.0063 kg/year in areas furthest away from roads.

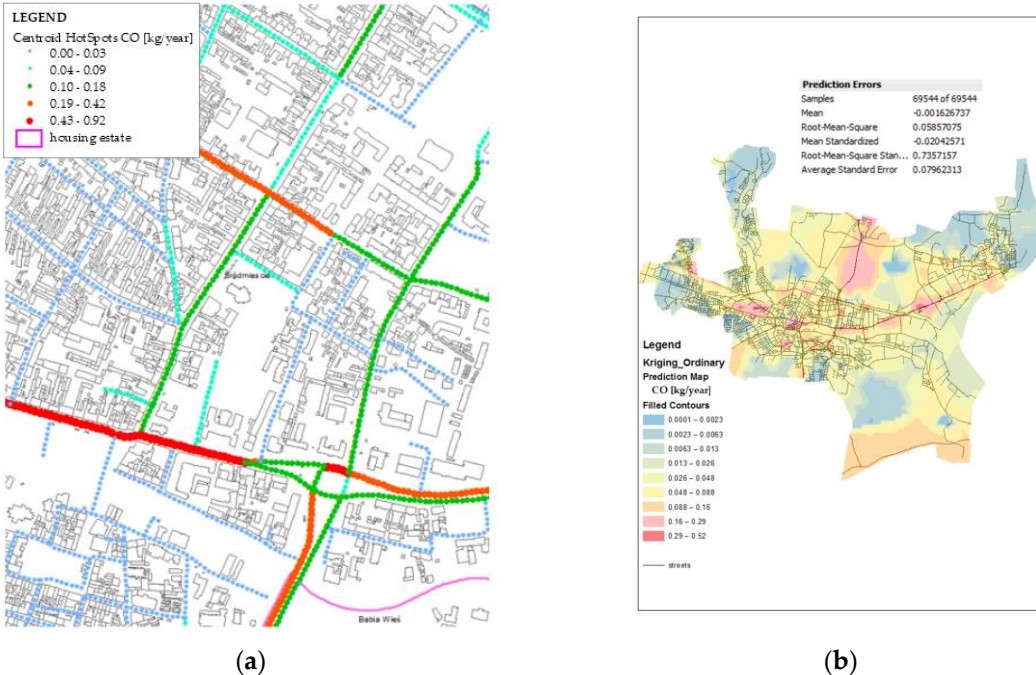

(**a**)                                                                 (**b**)

**Figure 5.** (**a**) Centroid hotspots of CO for each road segment; (**b**) prediction map of pollution of air with carbon monoxide for Bydgoszcz in 2018.

The aim of the studies was to determine the size of areas at risk of carbon monoxide (CO) pollution derived from road traffic along with determining the amount of people exposed to excessive levels of CO. Firstly, the total concentration of CO (mg/m$^3$/year) was calculated for the city. Accounting for the road network, the amount of CO at the level of 3591.90 kg, and the area of the city at 173,251,487.66 m$^2$, it was calculated that the CO concentration level for Bydgoszcz in 2018 was 5.18 mg/m$^3$/year. At the same time, in accordance with the legal provisions, Bydgoszcz lies in Class A of the Air Quality Standards, which confirms the good quality of air. Next, the average concentration of CO for individual housing estates was calculated, where calculation analyses were referred to the surface area of the estates and amount of CO (kg/year) emitted by motor vehicles driving down neighborhood streets. The results have been presented in Figure 6. As can be seen, the level of concentration of carbon monoxide for individual housing estates varies, ranging from 0.08 mg/m$^3$/year (Powiśle estate) to 35.70 mg/m$^3$/year (Wzgórze Wolności estate).

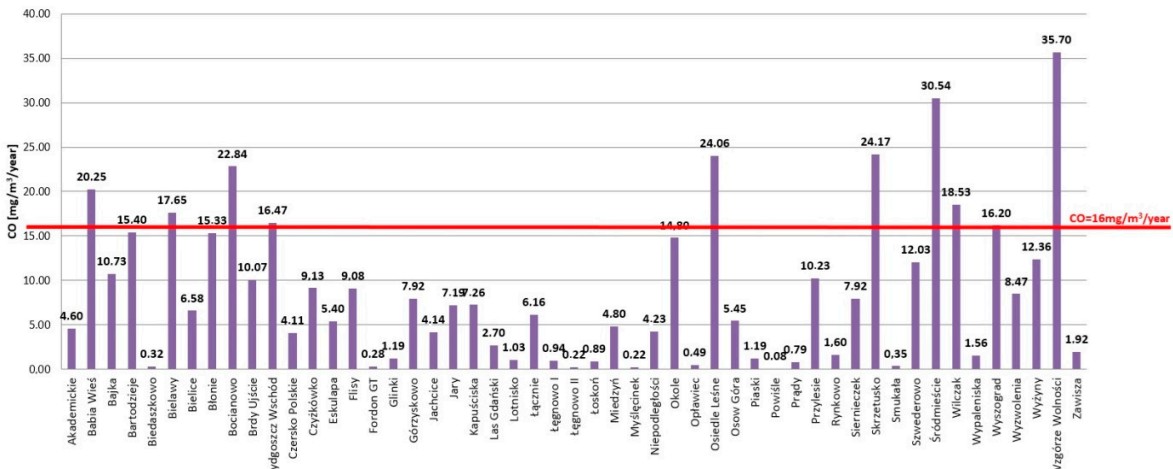

**Figure 6.** Level of CO concentration (mg/m$^3$/year) for all neighborhoods of the city of Bydgoszcz.

Detailed studies were carried out for housing estates where limit levels of carbon monoxide were found to have been exceeded, including ten housing estates located by transit roads, which are characterized by a high level of urbanization. Ten of the selected estates are located in the central part of the city, while one is 17 km east (Wyszogród estate) (Figure 7). An area with a total surface of P = 15.55 km$^2$ was analyzed (including $P_{min}$ = 0.62 km$^2$—Wilczak estate, $P_{max}$ = 3.79 km$^2$—Bydgoszcz Wschód estate) (Figure 8a). Residential areas of the selected estates are formed by single and multi-family development, including low buildings up to five stories high, as well as high 10-storey buildings, which are inhabited by over 80,000 people (Figure 8b). In addition to residential areas, there are commercial, industrial, and green areas (e.g., city parks) in the selected estates. The average concentration of CO for the selected estates was as much as 22.58 mg/m$^3$/year, including the maximum level noted for Wzgórze Wolności estate (35.70 mg/m$^3$/year) and the minimal level for Wyszogród estate (16.20 mg/m$^3$/year) (Table 1).

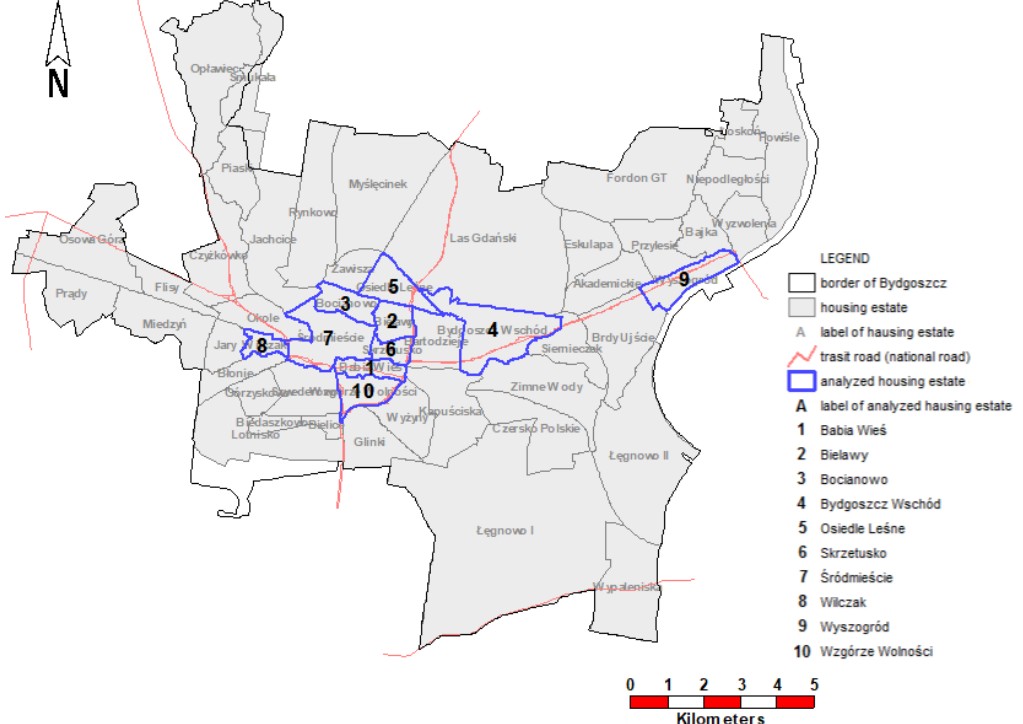

**Figure 7.** Map of Bydgoszcz with selected estates for which the limit level of CO (mg/m$^3$/year) concentration was exceeded.

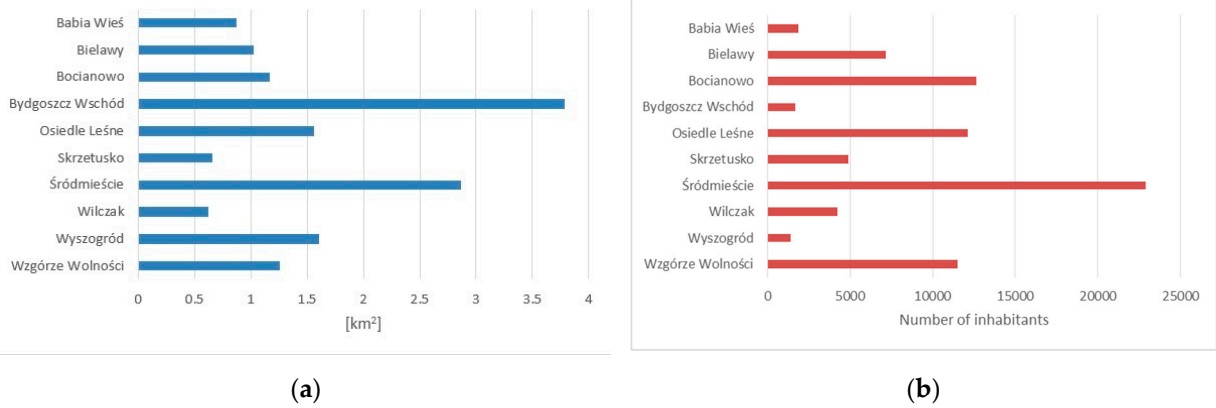

**Figure 8.** Selected estates: (**a**) area of the estate; (**b**) the number of inhabitants.

**Table 1.** List of estates for which the limit level of CO concentration (mg/m$^3$/year) was exceeded—limit value for CO = 16 mg/m$^3$/year.

| Name of Housing Estate | Surface Area of the Estate (m$^2$) | Volume for 4 m (m$^3$) | CO (kg/Year) | CO (mg/m$^3$/Year) |
|---|---|---|---|---|
| Babia Wieś | 873,433.62 | 3,493,734.49 | 70.76 | 20.25 |
| Bielawy | 1,030,997.99 | 4,123,991.98 | 72.80 | 17.65 |
| Bocianowo | 1,171,383.61 | 4,685,534.45 | 107.00 | 22.84 |
| Bydgoszcz Wschód | 3,786,232.28 | 15,144,929.13 | 249.50 | 16.47 |
| Osiedle Leśne | 1,559,725.76 | 6,238,903.03 | 150.12 | 24.06 |
| Skrzetusko | 656,878.19 | 2,627,512.78 | 63.52 | 24.17 |
| Śródmieście | 2,867,860.43 | 11,471,441.72 | 350.38 | 30.54 |
| Wilczak | 616,740.45 | 2,466,961.81 | 45.71 | 18.53 |
| Wyszogród | 1,606,134.04 | 6,424,536.16 | 104.09 | 16.20 |
| Wzgórze Wolności | 1,259,752.29 | 5,039,009.15 | 179.90 | 35.70 |
| In total | 15,429,138.67 | 61,716,554.70 | 1393.76 | 22.58 |

Next, the surface areas of the selected estates that were most at risk of carbon monoxide pollution were determined. The introduction of this step was the result of the analyses of spatial data presented on the prediction map (Figure 5b), where it was shown that the highest amount of CO (kg/year) is concentrated along roads, and along with increasing distance from roads, this level decreases significantly, oftentimes assuming very low values in central parts of estates. This, in turn, may signify good air quality, where the level of carbon monoxide is below the acceptable value. At the same time, when using Intergraph GeoMedia tools as well as prediction maps, the area of the selected estates was divided into two zones of carbon monoxide effect, distinguishing:

- areas located outside of the carbon monoxide concentration danger zone < 16 mg/m$^3$/year;
- areas within the carbon monoxide concentration danger zone > 16 mg/m$^3$/year.

Areas for which the level of CO on the prediction map was above 0.088 kg/year were included in the danger zone of carbon monoxide pollution. As can be seen in Figure 9, danger zones are concentrated around roads. Along with an increase in the amount of generated carbon monoxide, the danger zones covers a larger area of the estate, which together amounts to 33.36% of the area of the selected estates. The danger level, however, differs depending on the selected estate, its size spatial structure, and proximity of transit roads and ranges from 12.01% (Osiedle Leśne estate—Figure 10) to 57.89% (Wyszogród estate—Figure 11)—Table 2. The above means that, analyzing 10 selected estates for which the average level of CO concentration exceeded the value of 16 mg/m$^3$, as much as 66.64% of the area has good air quality when it comes to carbon monoxide (below the limit value). A sample prediction map as well as a map of the range of occurrence of zones affected by carbon monoxide along

with the location of residential buildings at risk of excessive levels of CO (danger zone—red color) are presented in Figures 10–16.

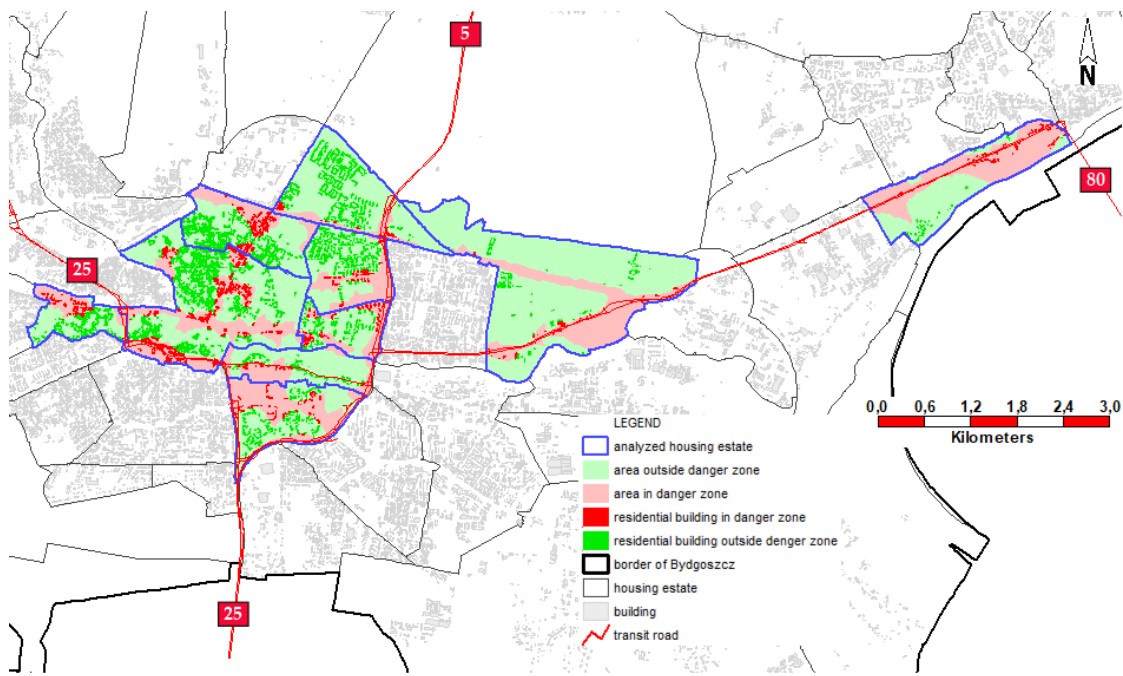

**Figure 9.** The spatial extent of zones of carbon monoxide pollution for selected housing estates.

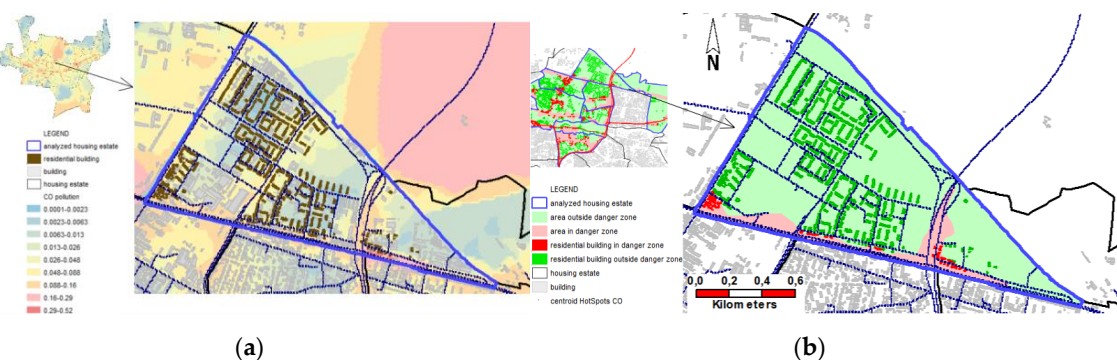

(**a**)　　　　　　　　　　　　　　　　　(**b**)

**Figure 10.** Osiedle Leśne estate: (**a**) prediction map of CO pollution (kg/year); (**b**) spatial scope of zones of carbon monoxide influence with residential buildings at risk.

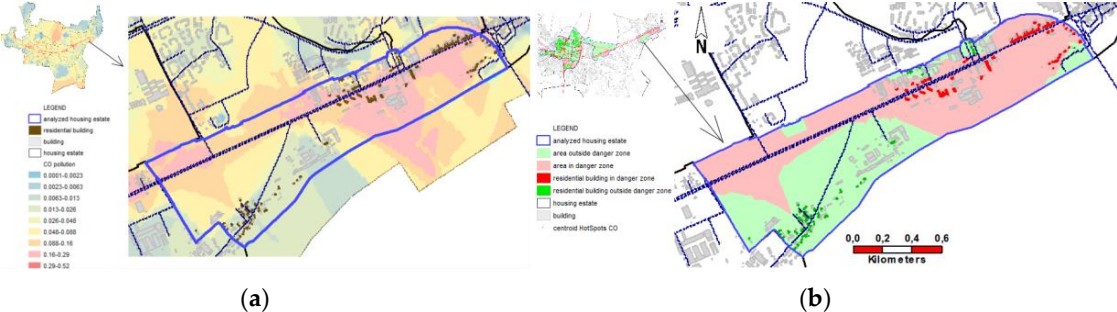

(**a**)　　　　　　　　　　　　　　　　　(**b**)

**Figure 11.** Wyszogród estate: (**a**) prediction map of CO pollution (kg/year); (**b**) spatial scope of zones of carbon monoxide influence with residential buildings at risk.

**Table 2.** Areas of selected housing estates lying in the danger zone of carbon monoxide pollution.

| Name of Housing | Surface Area of Estate (m²) | Areas in the Danger Zone of Carbon Monoxide Pollution | | | | Area Subjected to Excessive Levels of CO (%) |
|---|---|---|---|---|---|---|
| | | Surface Area (m²) | Volume (m³) | CO (kg/year) | CO (mg/m³/year) | |
| Babia Wieś | 873,433.62 | 288,914.00 | 1,155,656.00 | 52.22 | 45.19 | 33.08 |
| Bielawy | 1,030,997.99 | 375,341.70 | 1,501,366.80 | 65.77 | 43.81 | 36.41 |
| Bocianowo | 1,171,383.61 | 339,393.10 | 1,357,572.40 | 73.92 | 54.45 | 28.97 |
| Bydgoszcz Wschód | 3,786,232.28 | 945,992.50 | 3,783,970.00 | 206.87 | 54.67 | 24.99 |
| Osiedle Leśne | 1,559,725.76 | 187,390.30 | 749,561.20 | 101.34 | 135.20 | 12.01 |
| Skrzetusko | 656,878.19 | 273,818.70 | 1,095,274.80 | 56.85 | 51.90 | 41.68 |
| Śródmieście | 2,867,860.43 | 959,421.70 | 3,837,686.80 | 242.64 | 63.22 | 33.45 |
| Wilczak | 616,740.45 | 204,476.20 | 817,904.80 | 29.55 | 36.13 | 33.15 |
| Wyszogród | 1,606,134.04 | 929,762.10 | 3,719,048.40 | 99.81 | 26.84 | 57.89 |
| Wzgórze Wolności | 1,259,752.29 | 641,881.70 | 2,567,526.80 | 140.49 | 54.72 | 50.95 |
| In total | 15,429,138.67 | 5,146,392.00 | 20,585,568.00 | 1069.46 | 51.95 | 33.36 |

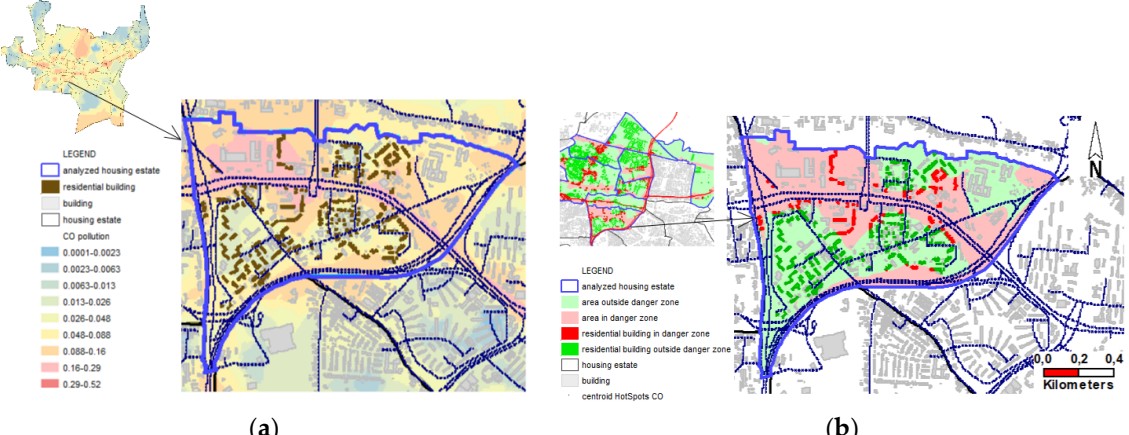

**Figure 12.** Wzgórze Wolności estate: (**a**) prediction map of CO pollution (kg/year); (**b**) spatial scope of zones of carbon monoxide influence with residential buildings at risk.

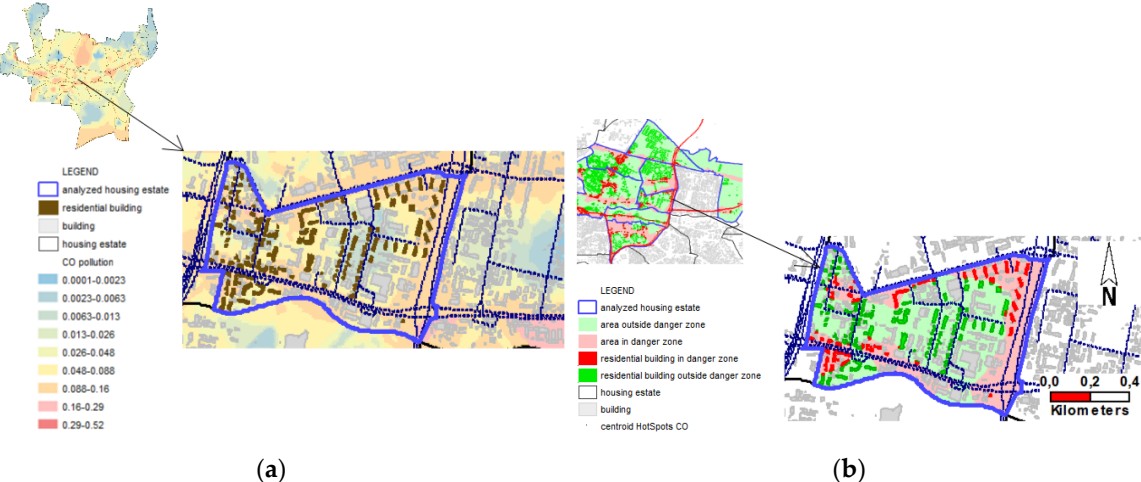

**Figure 13.** Skrzetusko estate: (**a**) prediction map of CO pollution (kg/year); (**b**) spatial scope of zones of carbon monoxide influence with residential buildings at risk.

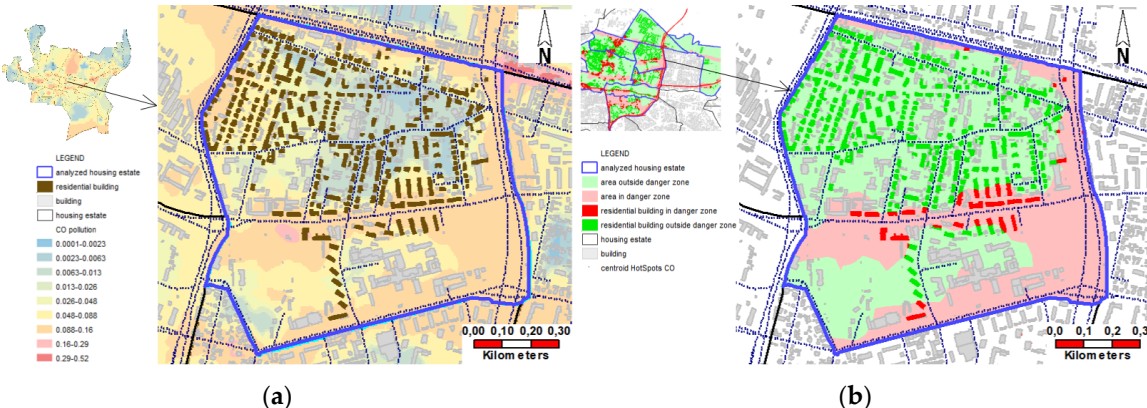

**Figure 14.** Bielawy estate: (**a**) prediction map of CO pollution (kg/year); (**b**) spatial scope of zones of carbon monoxide influence with residential buildings at risk.

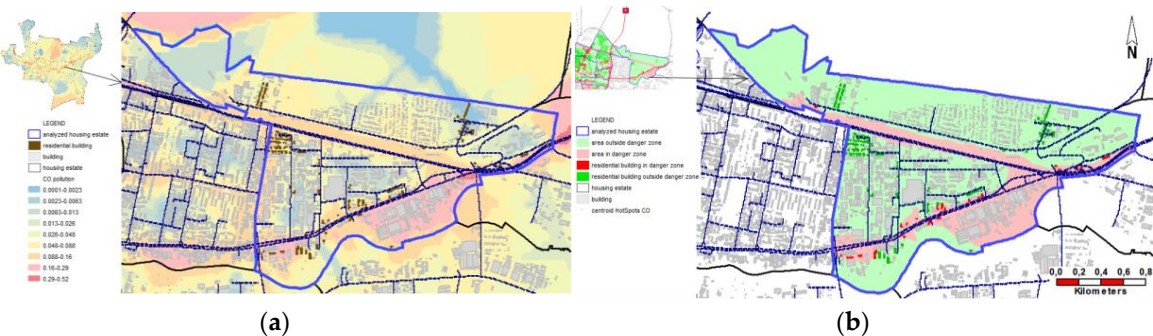

**Figure 15.** Bydgoszcz Wschód estate: (**a**) prediction map of CO pollution (kg/year); (**b**) spatial scope of zones of carbon monoxide influence with residential buildings at risk.

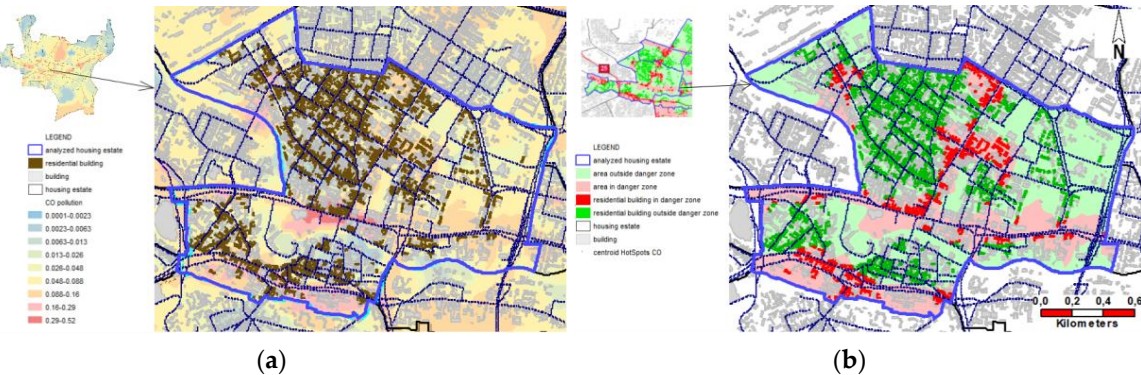

**Figure 16.** Śródmieście estate: (**a**) prediction map of CO pollution (kg/year); (**b**) spatial scope of zones of carbon monoxide influence with residential buildings at risk.

A total of 80,421 inhabitants reside in the selected housing estates. After assigning the inhabitants to the existing buildings included in one of two carbon monoxide pollution zones, it was concluded that nearly 45% of the inhabitants are at risk of excessive levels of CO (people residing in buildings lying in the danger zone of carbon monoxide pollution—Table 3). The highest value was noted for the Wzgórze Wolności estate—Figure 12. The above results from a few issues. Firstly, the boundary of the estate is formed by national roads No. 5 and 25, which, together with the remaining roads in the estate, emit a large amount of carbon monoxide, giving an average value at the level of 35.70 mg/m$^3$/year (see Table 1). Secondly, the location of high multi-family housing along the national roads led to a situation where nearly 84% of the inhabitants of Wzgórze Wolności estate are at risk of excessive levels of CO

(Table 3). An equally high amount of residents at risk is found in the Wyszogród estate. The location of Road No. 80 within the Wyszogród estate emits high amounts of CO (Table 2), and the concentration of multi-story residential development along this road led to 82% of the inhabitants of this estate being subjected to excessive levels of CO (Table 3). A similar situation pertains to the Skrzetusko estate—Figure 13. Somewhat better results were obtained in the areas of the Bielawy, Bydgoszcz Wschód, and Śródmieście estates (Figures 14–16), where approximately 40% of the inhabitants are at risk of excessive levels of CO. The best air quality in terms of carbon monoxide is in the Osiedle Leśne estate, which carries over into a low percentage of the population at risk (merely 4.71%)—see Table 3.

**Table 3.** Population of selected housing estates residing in the danger zone of carbon monoxide pollution.

| Name of Housing Estate | Estates in Total | | Area in Danger Zone of CO Contamination | | Population at Risk of Excessive Levels of CO (%) |
| --- | --- | --- | --- | --- | --- |
| | Number or Residential Buildings | Number of Inhabitants | Number of Residential Buildings | Number of Inhabitants | |
| Babia Wieś | 178 | 1883 | 64 | 739 | 39.25 |
| Bielawy | 463 | 7153 | 65 | 3412 | 47.70 |
| Bocianowo | 755 | 12,661 | 238 | 3388 | 26.76 |
| Bydgoszcz Wschód | 200 | 1674 | 46 | 732 | 43.73 |
| Osiedle Leśne | 357 | 12,119 | 71 | 571 | 4.71 |
| Skrzetusko | 237 | 4901 | 123 | 4037 | 82.37 |
| Śródmieście | 1713 | 22,889 | 559 | 10254 | 44.80 |
| Wilczak | 503 | 4218 | 135 | 1662 | 39.40 |
| Wyszogród | 157 | 1421 | 78 | 1162 | 81.77 |
| Wzgórze Wolności | 345 | 11,502 | 167 | 9622 | 83.66 |
| In total | 4908 | 80,421 | 1546 | 35579 | 44.24 |

## 4. Discussion

When analyzing the scientific works [45–60] as well as legal provisions [14], it can be stated that there is currently incomplete information regarding the air quality when it comes to carbon monoxide, which results from a few reasons:

- classes of air quality assessment are provided for the entire city in a uniform manner (one class is specified for the entire city) without an internal division accounting for the variability of sources of pollution, including the distance of the given areas from roads as well as changes in traffic intensity on individual road segments, which influences the level of carbon monoxide emitted from the road;
- classifying a city in a particular class is the result of the interpretation of measurements that were carried out in a continuous manner at measurement stations with installed sensors, without accounting for the propagation of pollutants in the air within the area of the city. Oftentimes, one or (sometimes) two measurement stations operate in the area of a single zone, the results of which are then referred to the entire surface area of the city. As shown in Figure 17, the level of the annual average CO concentration recorded in the two existing air quality monitoring stations for the city of Bydgoszcz in 2018 did not exceed 0.50 mg/m$^3$/year [54] (p. 77). At present, this value is the basis for the annual assessment of air quality in the city. The above stations are situated in the Śródmieście estate in Bydgoszcz (Figure 1c). Much higher CO concentration values were obtained with the use of geostatistical modeling for the Śródmieście estate. This level was over 30 mg/m$^3$/year;
- mathematical modeling of transport and the changes of substances in the air do not account for all air pollutants that were indicated in the European Union directive [14]. Advanced research methods are applied for the assessment of other air pollutants [50–53] and often the level of carbon monoxide derived from motor vehicle traffic is often overlooked, though a model based on the maximum concentration of CO for the given terrain accounting for dispersion in the vertical and

horizontal direction was developed in a publication [55]. The Hadipour team also accounted for the distance of residential areas from the urban transportation network and based the modeling of carbon monoxide pollution on road traffic data.

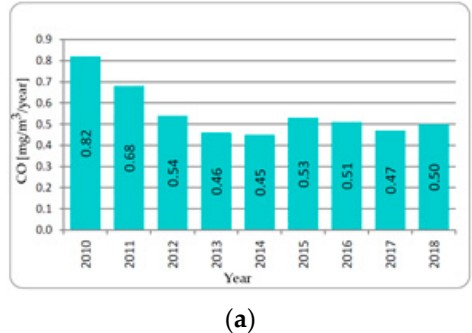

(**a**)

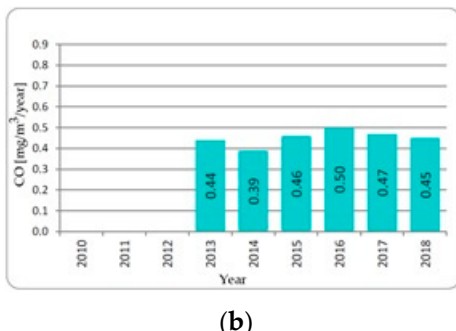

(**b**)

**Figure 17.** Changes in the level of CO concentration (mg/m³/year) in the period from 2010–2018 recorded by the existing air quality monitoring stations in Bydgoszcz (Śródmieście housing estate): (**a**) MS1—Pl. Poznański Street; (**b**) MS2—Warszawska Street.

The studies conducted by the authors indicate that only the detailed assessment of air quality based on geostatistical analyses accounting for the spatial structure of terrain and the variability of sources of pollution (dynamic of motor vehicle traffic) can give reliable information on the level of pollutions in the city. A similar view was expressed by [10,11,30]. The application of an integrated research method connecting advanced GIS technologies with the COPERT emissions calculator allows for a very detailed means of selecting areas in danger of carbon monoxide pollution and, thus, the effective selection of measures aimed at improving air quality. In the opinion of the authors, drawing conclusions on air quality based on data obtained from sensors installed on measurement stations or remote sensing data as well as analyzing these data in reference to the zone (entire city) is insufficient and burdened with errors. This is confirmed by results obtained in Bydgoszcz. The level of total concentration of CO in air calculated for the city (zone) is much lower than the limit level (CO = 5.18 mg/m³/year), classifying the city as Class A, i.e., an area with good air quality. Results at the level of 5 mg/m³/year were also obtained in other European cities [8]. However, detailed studies carried out using modern geostatistical methods provided contrary results. This is seeing as how it turned out that part of the air in the city is polluted with carbon monoxide to a much greater degree, even at the level of 35.70 mg/m³/year, while ten housing estates are characterized by poor air quality and fall into Class C. Moreover, over the course of the studies it turned out that further limiting the area of the ten selected estates is necessary, as it was found that the proximity of a road where there are high quantities of carbon monoxide necessitates the distinguishing of areas in the danger zone (those found close to the roads) as well as those outside the danger zone (those found in the central parts of housing estates, far from roads with a high traffic intensity). Upon excluding areas with good air quality outside the danger zone from the analyses, it turned out that nearly 45% of inhabitants of ten housing estates are subjected to excessive levels of carbon monoxide.

As can therefore be seen, the full assessment of air quality in urban areas requires detailed studies, where geostatistical methods provide good results. This view has been confirmed by other researchers [56,57], who confirm that the dangers connected with pollution in cities are often the results of a lack of knowledge and the comprehensive analysis of data at the local level. The team of scientists in Portugal [10,11] confirmed, on the other hand, that increasing the precision of modeling air quality from road traffic is a key issue for management in the road transport sector. Detailed studies of urban areas with division into estates where excessive levels of air pollution with carbon monoxide caused by road transport may occur were not found in the analyzed literature. Although studies pertaining to CO pollution [27,58,59] do exist, they pertain mainly to models estimating air pollution as well as analyses of spatial data with the use of the dispersion model [60]. Research on the influence of carbon monoxide

air pollution on the condition of the environment without accounting for meteorological parameters was presented in the present work. It should be noted that the conducted studies are based on average annual CO concentration values and only indirectly take into account seasonal fluctuations through the AADT value. For this reason, the obtained results should be treated as preliminary. The study also did not consider potential high-risk areas, such as underground and multi-story car parks, road tunnels, and various other partially or completely enclosed microenvironments with insufficient ventilation. Taking the above factors into account will constitute future research directions. In addition, in future research, the authors plan to consider meteorological parameters, thus increasing the accuracy of the obtained results.

## 5. Conclusions

An integrated research method for assessing CO concentration as a result of road traffic and using the example of a large Polish city was developed in this article. Based on the results of studies, the following detailed conclusions can be presented:

- the total level of carbon monoxide concentration in the air for the city was 5.18 mg/m$^3$/year. In accordance with legal provisions, Bydgoszcz, as a zone subjected to the assessment of air quality, falls into Class A, signifying good air quality in terms of carbon monoxide pollution;
- calculation analyses carried out for each Bydgoszcz housing estate suggests that the average level of carbon monoxide concentration in the air is different for the individual estates and ranges from 0.08 mg/m$^3$ to 35.70 mg/m$^3$/year. Of the 51 analyzed housing estates, ten are subjected to excessive levels of CO. At the same time, ten Bydgoszcz housing estates fall into Class C, which signifies poor air quality where pollution with carbon monoxide is concerned;
- differences in the level of carbon monoxide concentration in the air in the individual estates result, above all, from the types of roads co-creating the transportation network of the estate;
- the internal division of estates into two zones of carbon monoxide pollution confirmed that the highest concentration of CO pertains to areas located in the direct proximity of roads with a high traffic volume (transit roads). The inside parts of housing estates, located far from these roads, are characterized by good air quality. Only 33.36% of the selected estates lie within the danger zone of carbon monoxide pollution, whereas the remaining area is characterized by good air quality;
- in the area of the selected estates, approximately 45% of the population is exposed to excessive levels of carbon monoxide. This percentage varies for the individual estates, ranging from 4.71% to as much as 83.66%. The situation results from a few issues. The first of these is the average level of carbon monoxide concentration in the air, while the second is the location of residential multi-family buildings in the direct proximity of roads with high traffic volume. Oftentimes, single-family housing (characterized by a small number of inhabitants) is located in the central parts of neighborhoods, away from busy roads. Multi-family housing (with a large number of inhabitants), on the other hand, is located in peripheral parts of the estates, in the direct proximity of roads generating large amounts of CO. In this case, a road has a dual nature. On one hand, it facilitates easy access, on the other, it becomes a source of air pollution and negatively influences the health of the inhabitants.

Due to the exceeding of CO's limit value in 10 housing estates in Bydgoszcz, it is proposed to introduce air quality monitoring stations for CO. The carried out studies show in what way geostatistical tools form the ArcGIS package can be used to map spatial variation of air pollution in a city. The obtained spatial details can be used to improve the estimates based on interpolation between terrain observations and prediction models. The maps of pollutants based on so-called geostatistical models using a digital model of the city along with traffic parameters allow for the ongoing analyses and assessment of the state of the environment. Modelling the level of air pollution makes it possible to assume immediate actions aimed at maintaining the Air Quality Standards [14]. The proposed method



can be applied by European countries in an effort to assess air quality and the successive planning of protective measures, in view of the binding legal provisions.

**Author Contributions:** J.K. and K.S. conceived, designed, and performed the research and analyzed the data. Both authors have read and agreed to the published version of the manuscript.

**Funding:** This work was funded by the Ministry of Science and Higher Education of Poland (No. BS-9/2018) for which the author would like to express his sincere gratitude.

**Acknowledgments:** We want to confirm the support given to us by the Bydgoszcz City Hall, which provided data related to road traffic.

**Conflicts of Interest:** The authors declare no conflict of interest.

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
