# Peer review of "Mapping Carbon Monoxide Pollution of Residential Areas in a Polish City"

_remotesensing, doi:10.3390/rs12182885_

Round 1
Reviewer 1 Report
The study is important to verify which neighborhoods the city population has the highest and lowest concentrations of carbon monoxide. This study that can point out, for example, where the city can grow in places that have a lower amount of CO, even for changes in traffic on the roads and especially to avoid more vehicles and inhabitants in areas where the concentration of CO is already high. In terms of public policy, it would be important to have air quality monitoring stations in these neighborhoods as the concentrations are higher to confirm that the values provided for in this article are really in line with this result of this work and also mainly to protect people.
What are the main economic activities in this city?
If there are industries, what is the number of industries, what is the size of the vehicular fleet and what are the main sources of carbon monoxide.
Are there air quality monitoring stations for CO in this city?
The concentration output of this model is daily, hourly or monthly?
If the exits output results in hourly you have to make a graph with hourly value in the X axis and value of the concentrations in the Y axis for the different neighborhoods, select at least 5 neighborhoods with the highest concentrations for this and another graph with lower concentrations , so you can know if the whole city follows a pattern of concentration in the daytime cycle or if the variation in different neighborhoods.
Identify at what time the concentrations of CO are higher and at what times the concentrations are lower. There is probably a difference between the different seasons and this was not addressed in the article.
If there is a monitoring station in the city, these results should be compared in a graph of the result of the hourly concentrations of the model with the result of the monitoring station, at least in some neighborhood where the study was carried out and has a monitoring station.
Does this model not consider the meteorological variables with temperature, wind speed and height of the boundary layer?
Author Response
Response to Reviewer 1 Comments
Point 1: What are the main economic activities in this city?
Response 1: The main types of economic activity were entered into the article (see I 142-148). Additionally, fig. 1b was corrected, where the city's main economic activities were added.
Point 2: If there are industries, what is the number of industries, what is the size of the vehicular fleet and what are the primary sources of carbon monoxide.
Response 2: The article has been supplemented with information about the industries and sources of carbon monoxide (see 142-145) resulting from road traffic intensity and vehicle number. (l. 157)
Point 3: Are there air quality monitoring stations for CO in this city?
Response 3: Yes. There are two CO air quality monitoring stations in Bydgoszcz. These are automatic, stationary stations, located in the ÅšródmieÅ›cie estate. The location of the monitoring stations is shown in Figure 1c. Information on existing stations was introduced into the work (L. 162-168). Besides, Figure 17 has been added with the values of the average annual CO levels recorded at two air quality monitoring stations over the years. The obtained results were analyzed and compared with the amounts recorded in measuring stations (L. 386-393). A conclusion was added regarding the proposal to locate new measuring stations in housing estates, exceeding the limit level of CO (L. 480-483). The literature on measuring stations has been added (see [53]).
Point 4: The concentration output of this model is daily, hourly, or monthly?
Response 4: In the article, the average annual CO value was calculated using the COPERT STREET LEVEL program, into which road traffic data was introduced, calculated based on the AADT equation, taking into account daily, weekly, and monthly fluctuations. (See eq.1)
Point 5: If the exits output results in hourly you have to make a graph with hourly value in the X axis and value of the concentrations in the Y axis for the different neighbourhoods, select at least 5 neighbourhoods with the highest concentrations for this and another graph with lower concentrations , so you can know if the whole city follows a pattern of concentration in the daytime cycle or if the variation in different neighbourhoods.
Response 5: Hourly values are not included in the article. Instead, CO pollution prediction maps were added for the highest and lowest concentrations in the city, shown in Figures 10-16.
Point 6: Identity at what time the concentrations of CO are higher and at what times the concentrations are lower. There is probably a difference between the different seasons, and this was not addressed in the article.
Response 6: The article additionally adds a critical remark regarding the lack of consideration of seasonality (L 441-444). Taking this phenomenon into account will guide future research.
Point 7: If there is a monitoring station in the city, these results should be compared in a graph of the result of the hourly concentrations of the model with the result of the monitoring station, at least in some neighbourhood where the study was carried out and has a monitoring station.
Response 7: The information on existing stations was introduced into the article (L. 162-168). Figure 17 has also been added with the values of the average annual CO levels recorded at two air quality monitoring stations over the years. The obtained results were analyzed and compared with the values recorded in measuring stations (L. 386-393). A conclusion was added regarding the proposal to locate new measuring stations in housing estates, exceeding the limit level of CO (L. 480-483). The literature on measuring stations has been added (see [53]).
Point 8: Does this model not consider the meteorological variables with temperature, wind speed, and height of the boundary layer?
Response 8: Only the boundary layer height is included.
Reviewer 2 Report
I have had the opportunity to consider the submitted manuscript (remotesensing-897925) and present my own assessment of the suitability for publication in Remote Sensing.
The paper has merit in terms of the approach as it presents a methodology of CO outdoor mapping in a Polish city using interpolation algorithms and GIS features. The authors have done a good approach by synthesizing in a comprehensive matter various aspects related to the presentation of the topic.
After the reading of the manuscript, I have some suggestions to be considered:
- try to include (at least in discussion) some remote sensing resources (i.e. satellite information) that can be merged in your modeling approach (some data fusion techniques or other approaches that extract traffic information or aerosol presence). This is important since the readers of Remote Sensing are interested about remote sensing applications.
- If possible, more descriptive and comparative statistics with tables should be included regarding the variability of CO concentrations and comparisons between various microenvironments
- The discussion should be improved. Carbon monoxide levels have a close quantitative and temporal association with the levels of other primary exhaust pollutants such as nitrogen monoxide and volatile organic compounds. This association was not considered. The authors should locate on the maps potential high-risk areas such as underground and multistorey car parks, road tunnels, and various other partially or completely enclosed microenvironments with insufficient ventilation, because the levels of exhaust pollutants from combustion engines may be much higher than the common ambient levels in street canyons. Also, the comparison between indoor and outdoor microenvironments regarding potential exposure is missing. Seasonal fluctuations were not accounted for.
- Conclusions section should contain more specific information related to methodological recommendations of the authors regarding the use of geospatial modeling and potential future research. Now, the current information is a repetition of the results looking like a second abstract. Authors should synthesize the main findings of their research pointing out the practical outcomes and key limitations of the approach (topography and meteorology).
- English Language: The complete article needs to be reviewed and revised as necessary for style and punctuation). I have provided some suggestions but the style should be re-checked by the authors several times before re-submission and the support of a native English speaker would be a plus. This will increase its scientific value and readability.
I have provided a non-exhaustive list of suggestions:
Title:
Mapping Carbon Monoxide Pollution of Residential Areas in a Polish City
Abstract:
L12 Why so-called geostatistical models? Delete so-called
L18 [kg/unit of time]?
L22 Is it 8 hours interval?
L25 Maybe “limit value” is a better-accepted term than “permissible level” (check throughout the manuscript see also L61)
L30 to improve estimated concentrations…
Introduction
Some important references are missing – WHO is a reliable source of information on CO e.g. https://www.euro.who.int/__data/assets/pdf_file/0020/123059/AQG2ndEd_5_5carbonmonoxide.PDF
IMPORTANT! the fixed-site monitoring data may reflect population exposures somewhat better at longer averaging times such as 8 hours. The authors should consider this interval and adjust their reported modeled concentrations.
Furthermore, they should point out that the concentration is generally highest at the leeward side of the “street canyon”, and there is a sharp decline in the concentration from pavement to rooftop level. The carbon monoxide levels are highest in personal cars, the mean concentrations being 2–5 times the levels measured in the streets. These particularities should be discussed more in the manuscript when presenting results.
L46 -nitrogen oxides (NOx).
L54-56 Reformulate for better readability.
L70 Avoid the use of doesn’t in a scientific paper – use does not
L77 repair measures – mitigation and remediation measures
L79 on the basis of – based on
Reformulate the amount of which is insufficient to fully recognize the level of pollution in cities
L83 air atmosphere - atmospheric air
L85 replace impossible with difficult
L87 in space and time
I suggest deleting the footnotes since you have pointed out the required references.
L90 Avoid verbose – e.g. Geostatistical tools make it possible to solve various… Geostatistical tools solve various…
L100 The authors analyzed
L115 reference or website required
L118 …, lying replace with located (attention to the use of comma throughout the manuscript)
L121 prognostic maps – prediction maps
L127 typo – concertation (suggestion: read the manuscript several times before resubmission)
Material and methods
L135 The city of Bydgoszcz, which is located in the north-central part of Poland was selected for the studies. – Did you have the possibility to choose from several cities? I think that “The studies were performed in the city of Bydgoszcz, which is located in the north-central part of Poland.” is more appropriate.
L136-138 Reformulate since it is unclear
L147 …Poland, lying – Poland, being situated…
L159 Reformulate caption (use of against…)
L162 Reformulate in a simple manner e.g. The methodology involved the following steps:
Use semicolon at the end of each bullet and start the line without a capital letter
Generating a prediction map of CO [kg] pollution - it is still unclear – specify the unit of time
L175-177 require reformulation
L184 the given road segment
L185 Try to maintain consistency (use past tense throughout the section) Two types of measurement segments were considered…
data can be obtained indirectly from the PPT Visum software (provide a reference or a website for this software)
L194 start with new paragraph i.e., The analyses…
L213 Example of carbon monoxide calculations
L225 typo coordinate
Check the Figure 6 b and translate in English all terms
Provide in the caption of table 1 the limit value for CO as reference to the data presented.
L409 typo concertation
Author Response
Response to Reviewer 2 Comments
Point 1: try to include (at least in discussion) some remote sensing resources (i.e., satellite information) that can be merged in your modeling approach (some data fusion techniques or other strategies that extract traffic information or aerosol presence). It is important since the readers of Remote Sensing are interested in remote sensing applications.
Response 1: The use of satellite remote sensing (e.g., the Copernicus Sentinel-5P satellite placed in Earth orbit on October 13, 2017 [26] allows to obtain rough data on air pollution in the form of trace gases such as carbon monoxide, nitrogen dioxide and ozone with a resolution of 7 x 3.5 km. It enables a preliminary assessment of air pollution in individual cities (L.90-93).
Point 2: If possible, more descriptive and comparative statistics with tables should be included regarding the variability of CO concentrations and comparisons between various microenvironments
Response 2: CO pollution prediction maps for seven different housing estates were added to the article; the results are shown in Fig. 10-16.
Point 3: The discussion should be improved. Carbon monoxide levels have a close quantitative and temporal association with other primary exhaust pollutants such as nitrogen monoxide and volatile organic compounds. This association was not considered. The authors should locate on the maps potential high-risk areas such as underground and multistorey car parks, road tunnels, and various other partially or completely enclosed microenvironments with insufficient ventilation, because the levels of exhaust pollutants from combustion engines may be much higher than the common ambient levels in street canyons. Also, the comparison between indoor and outdoor microenvironments regarding potential exposure is missing. Seasonal fluctuations were not accounted for.
Response 3: Additionally, a critical remark was added regarding the lack of consideration of the research's indicated issues (L 441-447). Taking these phenomena into account will be the direction of future research. In the article, the average annual CO value was calculated based on the AADT equation. Thus, the presented CO value indirectly considers daily, weekly and monthly fluctuations (see eq, 1). In the AADT formula, road traffic intensity gives changes in months and weekly (working days, non-working days) and traffic peaks (L. 205-209). Hourly values were not taken into account in the work.
Point 4: Conclusions section should contain more specific information related to methodological recommendations of the authors regarding the use of geospatial modeling and potential future research. Now, the current information is a repetition of the results looking like a second abstract. Authors should synthesize the main findings of their research, pointing out the practical outcomes and key limitations of the approach (topography and meteorology).
Response 4: In the Conclusions section, apart from the general results (bullet points 1 and 2), a detailed analysis of the spatial conditions of the studied area and the influence of the spatial structure were introduced. The model considers the height of the boundary layer but does not take into account meteorological variables, i.e., temperature and wind speed. In future research, the authors will prepare maps of pollutants based on a numerical 3D terrain model. Proposals for locating CO monitoring stations (L 480-483) in housing estates with exceeded CO permissible levels have been added to the applications.
Point 5: English Language: The complete article needs to be reviewed and revised as necessary for style and punctuation). I have provided some suggestions but the style should be re-checked by the authors several times before re-submission and the support of a native English speaker would be a plus. This will increase its scientific value and readability.
Response 5: The entire article has been revised for style and punctuation using Grammarly Premium in Chrome and a native speaker.
Point 6: List of suggestions
Response 6: All comments of the reviewer were taken into account. The text has also been improved in figures and tables.
Title
Point 7:
Response 7: As suggested by the reviewer, the title of the article was changed to: Mapping Carbon Monoxide Pollution of Residential Areas in a Polish City
Abstract:
Point 8:
Response 8:
L12 Why so-called geostatistical models? Delete so-called
Changed
L18 [kg/unit of time]?
Changed to [kg/year]
L25 Maybe "limit value" is a better-accepted term than "permissible level" (check throughout the manuscript see also L61)
Changed to limit value
L30 to improve estimated concentration…
Changed
Introduction
Point 9:
IMPORTANT! the fixed-site monitoring data may reflect population exposures somewhat better at longer averaging times such as 8 hours. The authors should consider this interval and adjust their reported modeled concentrations.
Furthermore, they should point out that the concentration is generally highest at the leeward side of the "street canyon", and there is a sharp decline in the concentration from pavement to rooftop level. The carbon monoxide levels are highest in personal cars, the mean concentrations being 2–5 times the levels measured in the streets. These particularities should be discussed more in the manuscript when presenting results.
Response 9: Additionally, a critical remark was added regarding the lack of consideration of the research's indicated issues (L 441-447). Taking these phenomena into account will be the direction of future research.
Material and methods
Point 10:
L135
The city of Bydgoszcz, which is located in the north-central part of Poland was selected for the studies. – Did you have the possibility to choose from several cities? I think that "The studies were performed in the city of Bydgoszcz, which is located in the north-central part of Poland." is more appropriate.
Response 10:changed to "The studies were performed in the city of Bydgoszcz, which is located in the north-central part of Poland"
Point 11:
L162
Reformulate in a simple manner e.g. The methodology involved the following steps:
Response 11:changed to "The methodology involved the following steps:"
Point 12:
L185 Try to maintain consistency (use past tense throughout the section) Two types of measurement segments were considered…
data can be obtained indirectly from the PPT Visum software (provide a reference or a website for this software)
Response12:
[34] https://www.ptvgroup.com/en/solutions/products/ptv-visum/ (accessed on May 10, 2020).
Besides, the remaining detailed comments of the reviewer regarding style and grammatical errors were directly included in the article.
Reviewer 3 Report
see attachment

Author Response
Response to Reviewer 3 Comments
Point 1:
Research Methodology and Data
I suggest the ballpoints are made into a figure to describe the process.
Response 1: The remark was taken into account, a diagram presenting the individual stages of mapping carbon monoxide pollution with the use of geoinformation data was added to the work (Fig. 2. L 193)
Point 2: I want to know more about the datasets used.
Response 2: the datasets used are:
- descriptive database of car traffic including average annual daily traffic volume for each segment of the street network
GIS database containing:
- a street network with descriptive attributes
- buildings and districts with descriptive features regarding the number of inhabitants
- database on amount of air pollutants included in the COPERT STREET LEVEL program
Point 3: My last comment refers to the final presentation of the results. I wish to see a discussion on how the results are related to previous research. For example, how does the more aggregated spatial analysis related to the findings in this paper?
Response 3: There are two CO air quality monitoring stations in Bydgoszcz. These are automatic, stationary stations located in the ÅšródmieÅ›cie estate. The location of the monitoring stations is shown in Figure 1c. Information on existing stations was introduced into the work (L. 162-168). A drawing presenting data on air pollution recorded in the existing monitoring stations is attached to the article - the results present average annual values ​​from 2010 - measuring station, ul. Pl. PoznaÅ„ski and since 2013 - the measuring station at ul. Warsaw. Figure 17 shows the values ​​of the average annual CO levels recorded at two air quality monitoring stations over the years. The obtained results were analyzed and compared with the measurement stations (L. 386-393). A conclusion was added regarding the proposal to locate new measurement stations in housing estates, with the CO limit exceeding (L. 480-483). The literature on measuring stations has been added (see [53]).
Point 4: My overall impression of the paper I have reviewed is that the authors provide essential insights into the field. However, there is more to do in the article when it comes to presenting the results. The authors make a point of the analysis of the sufficient scale covering the city. There is more material to be presented more accurately. This would strengthen the paper´s contribution.
Response 3: Information on the existing air monitoring stations has been added to the article, and the results obtained in the stations have been presented. In the part of the Discussion and Conclusions, there was content related to the comparative analysis of data recorded by stations and the results using geostatistical data. (L.162-168; Fig. 1c; L. 386-393; Fig. 17; L. 480-483) Critical comments were also added to the conclusions and directions of future research related to the topic (441-447).
Round 2
Reviewer 2 Report
The authors addressed my comments.